# Dose–Response of Creatine Supplementation on Cognitive Function in Healthy Young Adults

**DOI:** 10.3390/brainsci13091276

**Published:** 2023-09-01

**Authors:** Terence Moriarty, Kelsey Bourbeau, Katie Dorman, Lance Runyon, Noah Glaser, Jenna Brandt, Mallory Hoodjer, Scott C. Forbes, Darren G. Candow

**Affiliations:** 1Department of Kinesiology & Athletic Training, University of Northern Iowa, Cedar Falls, IA 50614, USA; kelsey.bourbeau@uni.edu (K.B.); dormank@uni.edu (K.D.); runyonl@uni.edu (L.R.); nglaser@uni.edu (N.G.); brandtjennaj@gmail.com (J.B.); hoodjmaa@uni.edu (M.H.); 2Department of Physical Education Studies, Brandon University, Brandon, MB R7A 6A9, Canada; forbess@brandonu.ca; 3Aging Muscle & Bone Health Laboratory, Faculty of Kinesiology & Health Studies, University of Regina, 3737 Wascana Parkway, Regina, SK S4S 0A2, Canada; darren.candow@uregina.ca

**Keywords:** cognitive function, creatine, functional near-infrared spectroscopy, prefrontal cortex, oxygenated hemoglobin

## Abstract

To determine if creatine (Cr) supplementation could influence cognitive performance and whether any changes were related to changes in prefrontal cortex (PFC) activation during such cognitive tasks, thirty (M = 11, F = 19) participants were evenly randomized to receive supplementation with Cr (CR10:10 g/day or CR20:20 g/day) or a placebo (PLA:10 g/day) for 6 weeks. Participants completed a cognitive test battery (processing speed, episodic memory, and attention) on two separate occasions prior to and following supplementation. Functional near-infrared spectroscopy (fNIRS) was used to measure PFC oxyhemoglobin (O_2_Hb) during the cognitive evaluation. A two-way repeated measures ANOVA was used to determine the differences between the groups and the timepoints for the cognitive performance scores and PFC O_2_Hb. In addition, a one-way ANOVA of % change was used to determine pre- and post-differences between the groups. Creatine (independent of dosage) had no significant effect on the measures of cognitive performance. There was a trend for decreased relative PFC O_2_Hb in the CR10 group versus the PLA group in the processing speed test (*p* = 0.06). Overall, six weeks of Cr supplementation at a moderate or high dose does not improve cognitive performance or change PFC activation in young adults.

## 1. Introduction

The adenosine triphosphate (ATP)–phosphocreatine (PCr) energy system plays an important role as a spatial and temporal energy buffer to aid in the functioning of the central nervous system [1]. Specifically, creatine (Cr) acts by aiding in the transfer of the N-phosphoryl group from PCr (generated in the mitochondria) to adenosine diphosphate (ADP), thereby rapidly resynthesizing ATP in the cytosol for use as free energy [2,3]. Almost the entirety of creatine is stored in skeletal muscle (95%). The vast majority of research has investigated the efficacy of creatine supplementation (primarily when combined with resistance training) on measures of muscle performance (i.e., strength, endurance) and muscle/lean mass in young, healthy populations [4]. In contrast to skeletal muscle, less than 5% of total body creatine stores are found in the brain [5]. The brain is very metabolically active, accounting for ~20% of the body’s energy consumption [6], and during mentally stimulating cognitive tasks, brain PCr decreases rapidly in order to maintain constant ATP levels [7]. The main enzyme involved in the production of ATP via the ATP-PCr energy system, creatine kinase (CK), has a brain-specific isoform (BB-CK) [8]. This suggests that Cr has the ability to supply energy and help with functioning in the central nervous system (CNS). In line, Cr supplementation may promote better cognitive function during conditions characterized by fluctuating ATP turnover, such as hypoxia [9], mild traumatic brain injury [5], sleep deprivation [10], and complex cognitive tasks in older adults [11].

The optimal creatine dosage needed to increase brain creatine levels is unclear. To date, only 11 studies have examined the relationship between creatine supplementation and brain creatine levels. Overall, creatine supplementation increased brain creatine levels by 3–10% [12]. However, there appears to be high variability in the responsiveness to creatine supplementation. Amongst the studies involving young, healthy adults, three studies found improvements in brain creatine levels due to creatine supplementation (~20 g/day for 7–28 days), whereas two studies found no effect in relation to creatine (~20 g/day for 7 days). In addition to dosage, there is also large heterogeneity across studies when it comes to supplementation duration, methods used to assess brain Cr levels (proton (H1-NMR) or phosphorus (P31-NMR) nuclear magnetic resonance spectroscopy) and the specific area of the brain assessed [13]. All in all, it appears that dosages of 20 g and over have shown the most consistent results in terms of increasing brain Cr levels in healthy populations.

Whether an increase in brain Cr levels from creatine supplementation leads to downstream improvements in central nervous system and cognitive functioning in young, healthy adults remains to be fully determined [5,14]. The effects of Cr supplementation on measures of cognitive performance in healthy non-vegan/non-vegetarian adults are varied, with some evidence indicating improved cognitive performance [15,16,17,18] while others indicate no benefits [19]. For example, Rawson and colleagues [19] found that supplementation with 0.03 g/kg/day of Cr for 6 weeks did not alter automated neuropsychological assessment metrics in male and female young adults, while Borchio et al. [15] reported increased performance in choice reaction time, selective attention and inhibition, and short-term memory following Cr supplementation of 20 g/day for 7 days in semi-professional, male mountain bikers. Since there are such inconsistencies between the quantity of dosages (ranging from ~2 to 20 g/day) and duration of dosing protocols (ranging from 5 days to 6 weeks) in the aforementioned studies in healthy young adults, investigating any such changes for the longest supplementation duration in a dose–response manner is warranted.

In addition to investigating whether Cr supplementation modulates cognitive performance, limited efforts have been made to explore the underlying mechanisms that may explain the relationship between increases in brain Cr and PCr and cognitive function. One such mechanism that may explain the downstream changes in cognitive performance is a Cr-induced activation change in the prefrontal cortex (PFC). To date, the PFC (left) has only been examined in one study following 5 days of Cr supplementation (8 g/day) in young healthy adults [18]. Since the PFC is the most well-developed area of the cerebral cortex and communicates extensively with the surrounding regions (premotor cortex, cerebellum, basal ganglia, thalamus, hippocampus), it is involved in various cognitive functions (e.g., attention, decision-making, prediction, and memory) [20]. During cognitively demanding tasks, there is an increase in PFC activation; as such, the PFC has become a common site of investigation for researchers examining cognitive performance [21]. To examine the hemodynamic mechanisms underlying cognitive performance, functional near-infrared spectroscopy (fNIRS) can be used to monitor changes in oxygenated (O_2_Hb) and deoxygenated hemoglobin (HHb) [22]. The advantages of using fNIRS over other neuroimaging techniques are that it is noninvasive, portable, tolerates motion artifacts well, and has good spatial localization. In theory, the PFC would experience an increase in O_2_Hb and a reduction in HHb as cognitive difficulty increases (indicating an increase in activation).

Therefore, the primary purpose was to determine if 6 weeks of Cr supplementation influences cognitive performance and whether any such changes were related to changes in PFC activation (measured via fNIRS changes in O_2_Hb) during such cognitive tasks. We further aimed to explore if moderate (10 g/day) or high dosing (20 g/day) confers different physiological or cognitive adaptations.

## 2. Materials and Methods

### 2.1. Experimental Design

This study was a double-blind, repeated measures study conducted in the Department of Kinesiology at the University of Northern Iowa (UNI). During the pre-testing visit, measurements included height, body mass, body composition via bioelectrical impedance analysis, Godin physical activity questionnaire, cognitive function tasks (while donning the fNIRS), and a maximal oxygen consumption test (VO_2_max). All of these tests were repeated at the post-testing visit except for height and the VO_2_max test. Following all pre-testing measurements, participants were randomized to one of three supplement groups (*n* = 10 in each group) and asked to ingest the supplement each day for 6 weeks. Testing was conducted at the same time of day for both visits (±2 h), and participants reported to the laboratory in a post-prandial state.

### 2.2. Participants

Thirty healthy participants (11 males, 19 females) with a mean age of 21 years (19–33 years range) volunteered to participate (Table 1). All participants completed a health questionnaire, and all procedures, discomforts, and risks were discussed before written informed consent was obtained. Participants were instructed to maintain similar habitual exercise and dietary behaviors for the duration of the study. Individuals who had previously ingested Cr monohydrate supplements were permitted to participate in the study if they had not ingested Cr within the previous 4 weeks. Participants were excluded from the study if they had pre-existing kidney disease or liver abnormalities. All study procedures were performed in the Exercise Physiology Laboratory at UNI, and the protocol (22-0083) was approved by the UNI Institutional Review Board for Human Subject Research.

### 2.3. Pre-Post-Testing Measures

Following written consent, the Godin Physical Activity questionnaire was administered, and each participant’s height and weight were measured, respectively. These measures were used to determine each participant’s body mass index (BMI). After, body fat percentage was estimated using bioelectrical impedance analysis (BIA) (InBody 720, Cerritos, CA, USA). The same researcher performed the BIA test for all participants. It is important to note that BIA is not the gold standard for body composition evaluation, and although it is not a primary outcome in this study, the BIA results depend on BIA-based predictive equations with oscillating values depending on the formula applied and device [23].

Following these initial measurements, cognitive performance was evaluated using select tests from the NIH Toolbox Fluid Cognition Battery via an iPad [24]. Specifically, the pattern comparison test was used to assess processing speed, the picture sequence memory test was used to assess episodic memory, and the dimensional change card sort test was used to assess executive function. In addition to the specific cognitive domains, this battery was chosen because it was originally developed to provide efficient and effective measures of global cognition (i.e., one’s ability to think, solve problems, and develop memories) in research settings [25]. Fully corrected T-scores, which adjust for age, sex, race/ethnicity, and educational background, were generated for each cognitive test. A T-score of 50 indicates performance at the average level based on the input demographic information of that participant. Total test administration required approximately 12–15 min.

Lastly, participants performed a VO_2_max test on a motorized treadmill (Precor Inc., Woodinville, WA, USA) while wearing a chest Polar heart rate monitor (V800, Polar Electro Inc., Woodbury, NY, USA). After a self-selected 5 min warm up, participants were connected to a metabolic cart (True One; Parvomedics, Sandy, UT, USA) for continuous measurement of oxygen consumption and carbon dioxide production. The metabolic cart was calibrated prior to each participant in accordance with the manufacturer’s guidelines. All maximal exercise tests were completed at 3% grade, and the initial running speed was selected to correspond with an intensity that could be maintained for at least 30 min. Following this, the speed was increased at a rate of 0.1 mph every 15 s and all tests were completed within 8–10 min [26,27]. Termination of the exercise test occurred upon the participant reaching maximal exertion or volitional cessation. That said, participants were asked to inform the researcher if they experienced any chest pain or dizziness. VO_2_max required two of the following four criteria to be achieved: respiratory exchange ratio (RER) > 1.15, within 10 bpm of age-predicted maximal heart rate, VO_2_plateau of <150 mL/min, or rating of perceived exertion (RPE) > 17. Maximal oxygen consumption was determined by the highest value achieved using 15 s averaging.

### 2.4. Supplementation and Randomization

Cr (Creapure^®^ AlzChem, Trostberg GmbH, Trostberg, Germany) and PLA (Globe^®^ Plus 10 DE Maltodextrin, Univar, Richmond, BC, Canada) were in powder form. Both products were similar in taste, color (white), texture and appearance. One individual not involved in any other aspect of the study was responsible for randomizing participants into three groups to ensure that all participants and investigators remained blinded throughout the study: 10 g of Cr per day (CR10), 20 g of Cr per day (CR20), or 10 g of placebo per day (PLA). In addition, this individual was responsible for preparing participant study kits, which included their supplement for the duration of the 6 weeks. Participants were instructed to ingest the supplement in as many doses as needed for comfort, but all participants were instructed to ingest it with water. Adherence to the supplementation protocol was assessed by using a compliance log. Upon completion of the study, participants were asked whether they thought they were administered Cr, placebo, or did not know.

### 2.5. Fnirs

All fNIRS recordings during the pre- and post-testing cognitive tests were continuously captured using a dual-wavelength (760 and 850 nm), portable, noninvasive 8-channel system (Octamon, Artinis Medical Systems, Elst, The Netherlands). This device measures changes in O_2_Hb and HHb in the PFC, specifically covering the dorsolateral (DLPFC) and orbitofrontal (OFC) cortices in both hemispheres [28]. The headpiece location was identified by locating the naison site and placing the edge of the headpiece 2 cm about this point (approximately 1 cm above the brow line). The emitter to optode distance was 3.5 cm, resulting in a penetrated tissue of approximately 1.5 cm. The 8-channel system recorded data with a signal sampling of 10 Hz. Relative concentration changes (µM) were calculated with the manufacturer’s software (OxySoft version 3.2.72 ×64) using a modified Beer–Lambert law [29] and used as indicators of PFC oxygenation and activation. This software also adjusts the signals according to the differential pathlength factor (DPF) calculated by age [30]. The data were filtered with a 0.1 Hz low-pass filter to remove physiological noise (e.g., breathing and heart rate) and then averaged and analyzed in GraphPad Prism, version 9.4.1.

### 2.6. Diet

Participants completed a three-day food intake form to record all food and beverage consumption, including portion sizes, for three days (two weekdays and one weekend day). This form was administered during the first week and final week of supplementation and entered into MyFitnessPal to determine average total energy (kcal) and macronutrient intake over the three days. MyFitnessPal has been shown to have good relative validity, especially for energy [31].

### 2.7. Statistical Analyses

The sample size was estimated based on long-term memory data from McMorris et al. [32] with effect size d = 0.81 and conducted assuming a power of 0.80 and alpha level of 0.05. It was estimated that a total of 21 subjects would be needed to locate a significant effect. Similar sample sizes have been used to show reduced mental fatigue and reduced cerebral oxygenated hemoglobin (creatine = 12, placebo = 12) [18]. A 3 (CR10 vs. CR20 vs. PLA) × 2 (pre- and post-test time points) analysis of variance (ANOVA) with repeated measures was performed to determine differences between groups over time for cognitive performance scores and PFC O_2_Hb levels. A one-factor ANOVA was used to assess baseline data and absolute % change ((post − pre/pre) ×100). All results are expressed as means (standard deviation). Significance was set at an alpha level of *p* < 0.05. Statistical analyses were performed using GraphPad Prism, version 9.4.1.

## 3. Results

Baseline data are presented in Table 1. There were no differences between groups for any baseline measure. There were no changes in body mass (CR10: pre 70.0 ± 12.0, post 71.2 ± 12.0 kg; CR20: pre 76.2 ± 10.5, post 77.9 ± 10.3 kg; PLA: pre 75.5 ± 13.0, post 75.4 ± 12.5 kg), BMI (CR10: pre 23.4 ± 3.0, post 23.9 ± 3.0 kg/m^2^; CR20: pre 26.1 ± 2.9, post 26.7 ± 2.8 kg/m^2^; PLA: pre 25.5 ± 3.1, post 25.4 ± 2.9 kg/m^2^), body fat % (CR10: pre 24.2 ± 6.8, post 22.3 ± 7.1%; CR20: pre 26.1 ± 11.3, post 25.4 ± 11.8%; PLA: pre 21.4 ± 4.9, post 21.6 ± 4.9%), or Godin questionnaire physical activity data (CR10: pre 46 ± 30, post 55 ± 33; CR20: pre 57 ± 24, post 64 ± 24%; PLA: pre 44 ± 22, post 48 ± 29) over time. There were also no changes in absolute mean % change in cognitive performance across all three cognitive tasks or PFC O_2_HB during these tasks (Table 2). However, there was a trend towards reduced PFC O_2_HB during the first cognitive task, specifically for CR10 (*p* = 0.06) in comparison to PLA (Figure 1).

Adverse events reported included gastrointestinal symptoms such as gas (CR20: *n* = 2) and bloating or feeling full (CR20: *n* = 1). Following the intervention, participants were asked whether they thought they were administered Cr, PLA, or were unsure about what supplement they consumed. In the CR10 group, four participants correctly guessed they were consuming Cr, four guessed incorrectly, and two did not know. In the CR20 group, six participants correctly guessed they were consuming Cr, two guessed incorrectly, and two did not know. In the PLA group, three participants correctly guessed they were consuming a placebo, three guessed incorrectly, and four did not know.

## 4. Discussion

To the author’s knowledge, this is the first study to directly investigate the dose–response of 6 weeks of Cr supplementation on cognitive function and PFC O_2_Hb in healthy, young adults. It has been previously reported that Cr supplementation can increase brain Cr and phosphocreatine levels [13,33]. As a result, the increased brain Cr and phosphocreatine may lead to downstream altered cerebral O_2_Hb [18] and improved cognitive performance [15,16,17,18]. In the current study, fNIRS technology was utilized during three cognitive tasks to provide a global means of assessing cognitive function pre- and post- 6 weeks of varying dosages (10 g/day and 20 g/day) of Cr supplementation in healthy, young adults.

Contrary to the positive results of previous studies, our findings were unable to demonstrate any positive effects of moderate or high Cr dosage on processing speed, working memory, or executive function performance. Our findings are in line with the results of Rawson and colleagues [19], who report that there were no significant differences in a comprehensive battery of neurocognitive tests (including simple reaction time, code substitution, code substitution delayed, logical reasoning symbolic, mathematical processing, running memory, and Sternberg memory recall) following 6 weeks of Cr (0.03 g/kg/day) or PLA supplementation in young adults (six males and five females; 21.0 ± 2.1 years). Similarly, Merege-Filho et al. [34] also revealed that Cr supplementation of 0.3 g/kg/day divided equally into four doses for 7 days did not alter cognitive function or brain Cr levels in healthy male and female children (10–12 years). From a more longitudinal perspective, Alves et al. [35] found that Cr supplementation (20 g/day divided into equal doses for the first 5 days, followed by a single dose of 5 g/day for 24 weeks) did not influence cognitive performance in a variety of cognitive performance tasks. It is important to note that this study was conducted among older women (range 60 to 80 years), and older adults may respond differently to younger adults [36].

The supplementation protocol used in the current study (10 g/day and 20 g/day for 6 weeks) was higher than the protocol of Rawson et al. [19] and Merege-Filho et al. [34], and although shorter in duration, it was also of greater dosage per day than Alves et al. [35]. Interestingly, the Cr dosages administered in the present study are higher than a number of studies that report cognitive benefits following Cr supplementation in healthy young men and women [15,16,17,18,37]. Since the brain is of much smaller mass than the amount of total body skeletal muscle and the fact that the muscle total Cr concentration can be increased by ~20% in as little as 6 days with 20 g/day [38], we speculated that the dose administered in the current study (10 g/day or 20 g/day for 6 weeks) would be adequate to increase brain Cr, but this may not be the case. It is possible that higher and longer dosing strategies are needed in young, healthy adults. Perhaps young, healthy adults have higher baseline brain Cr levels to begin with, providing a potential explanation for the lack of cognitive change in the present study. For example, it is well established that individuals with lower levels of muscle Cr yield the largest increases in response to supplementation. In line, it is possible that only those healthy individuals with very high dosages and longer-term dosing strategies or lower baseline brain Cr levels to begin with will experience large enough increases to elicit cognitive change [37,39]. However, a main limitation of our study was that brain creatine levels were not assessed, so we are unable to determine whether our creatine supplementation protocol raised brain creatine levels over time. One such study that assessed brain Cr levels (left dorsolateral PFC, left hippocampus, and occipital lobe) and cognitive performance (verbal learning and executive function) in healthy youths found that neither brain Cr levels nor cognitive performance were influenced by the Cr supplementation [34]. Therefore, if adequate resources exist in future studies, Cr content should be measured in specific brain areas involved in expected cognitive functions (e.g., measurement of PFC Cr content along with assessment of executive function), allowing for the mapping of brain Cr and cognitive performance in the context of Cr supplementation.

Recently, monitoring the PFC using fNIRS technology has been used more and more commonly in cognitive performance studies [40,41,42]. An increase in PFC oxygenation during mentally challenging tasks is generally thought to represent an increase in psychological effort due to an increase in neural and metabolic activity [43]. Although there was a trend for decreased PFC O_2_Hb following the 10 g/day Cr supplementation during the first cognitive task (processing speed), in the current study, there were no differences in PFC activation (including significant relative % change) from pre to post timepoints (Table 2). Although not significant, the trend toward decreased PFC O_2_Hb during the processing speed task does follow the general results of Watanabe et al. [18], who report that a dietary supplement of Cr (8 g/day for 5 days) resulted in a reduction in mental fatigue and a significant reduction in cerebral (optodes were placed over the left PFC) oxygenated hemoglobin measured via fNIRS during a unique serial calculation task (Uchida–Kraepelin test). The authors in this study suggest that this decrease is compatible with increased oxygen utilization in the brain. In addition, a decrease in oxygenation as a result of Cr supplementation may be due to a more efficient utilization of the immediate energy system due to increased brain Cr levels and a decreased reliance on aerobic metabolism as a means of generating ATP. It may be the case that a decrease in O_2_Hb illustrates a shift in ATP synthesis pathways that may speculatively reduce mental fatigue during the cognitive task.

Several possible reasons may explain the cognitive performance and PFC oxygenation discrepancies between previous literature and the current findings. Firstly, it may be that the dosage and/or duration of the current protocol was not adequate to elevate PFC Cr levels. Secondly, a thorough assessment of global cognition via three consecutive cognitive tasks was administered in our study and included processing speed, episodic memory, and executive function. Other research groups have used a single cognitive measure, such as inhibitory control, working memory, or executive function [18]. It may also be the case that a more complex set of cognitive tasks or ones that induce greater mental fatigue are needed to prompt additional brain activity in healthy young adults. Thirdly, perhaps having the participants complete all three cognitive tests back-to-back may have also influenced the cognitive performance and PFC O_2_Hb findings. Finally, as with any research study of Cr supplementation, there is the possibility that some participants did not respond to the supplementation due to higher-than-normal baseline brain Cr levels. That said, since we did not measure baseline and post-intervention brain Cr levels, it is difficult to interpret any differences that could have contributed to any such contradicting results.

In addition to the lack of brain creatine measures, this study had other limitations. The small sample size and short-duration supplementation protocol may have influenced our findings. Further, cognitive performance was assessed using three specific cognitive tasks, and oxygenation was assessed in a specific cortical area using a superficial tissue measurement. Therefore, our results may differ from other research studies that measure different cognitive domains, utilize more invasive methods (e.g., fMRI) of assessing cerebral activation, or simply measure other specific cortical areas during cognitive testing. Finally, although total protein was not matched in this investigation, the difference in protein intake is still challenging in terms of quantifying Cr content and is highly unlikely to have influenced the results. Future research should examine different subsets of healthy and clinical populations (e.g., elderly, sleep-deprived, dementia) with varying cognitive tests and measurement tools.

## 5. Conclusions

In conclusion, 6 weeks of moderate- (10 g/day) and high-dose (20 g/day) creatine supplementation does not improve cognitive performance or alter PFC O_2_Hb levels in healthy young adults. Future research should include larger sample sizes and longer interventions as well as focus on the precise neurophysiological mechanisms that may be responsible for any changes in cognitive function in varied healthy and clinical populations.

## Figures and Tables

**Figure 1 brainsci-13-01276-f001:**
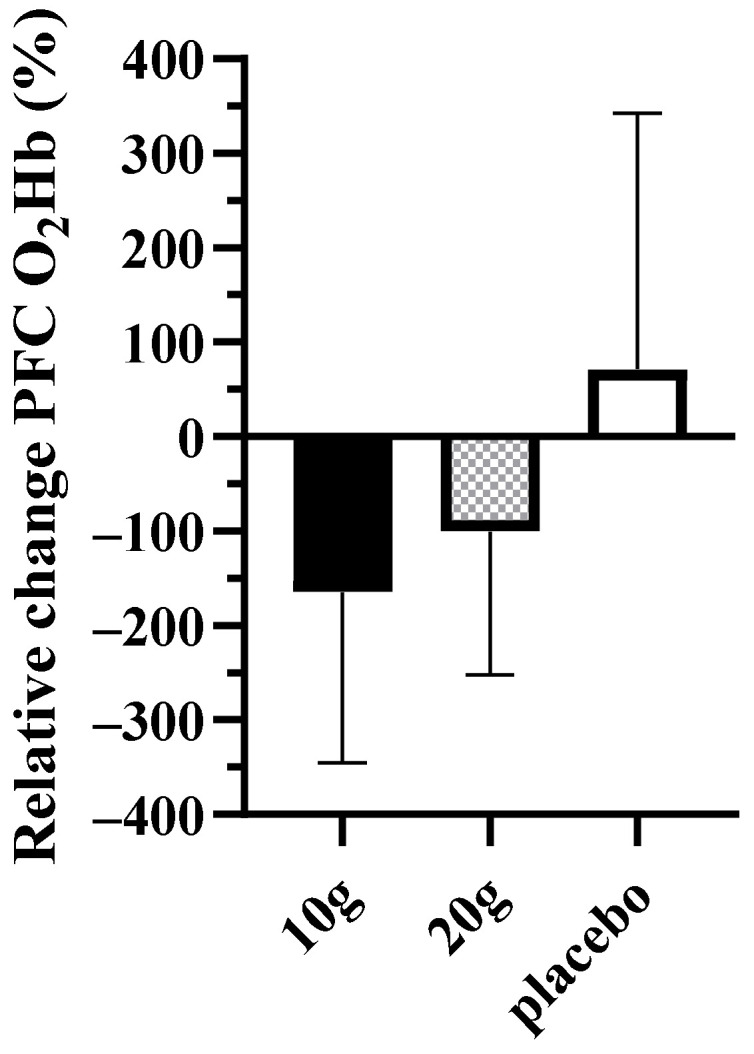
Relative change of PFC O_2_Hb responses during the first cognitive task (processing speed). O_2_Hb = oxyhemoglobin; PFC = prefrontal cortex.

**Table 1 brainsci-13-01276-t001:** Baseline characteristics.

Group (M/F)	CR10 (3/7)	CR20 (4/6)	PLA (4/6)	*p*-Value
Age (yrs)	21.8 (4.1)	20.4 (0.7)	20.8 (1.4)	0.42
Mass (kg)	70.0 (12.0)	76.2 (10.5)	75.5 (13.0)	0.46
Height (m)	1.72 (0.06)	1.71 (0.10)	1.72 (0.06)	0.90
Body Mass Index (kg/m^2^)	23.4 (3.0)	26.1 (2.9)	25.5 (3.1)	0.14
BF (%)	24.2 (6.8)	26.1 (11.3)	21.4 (4.9)	0.45
VO_2_max (mL·kg^−1^·min^−1^)	44.4 (11.1)	46.5 (8.7)	47.8 (5.6)	0.19
Godin questionnaire (physical activity)	46 (30)	57 (24)	44 (22)	0.51
Total calories (kcal/day)	2088 (516)	2143 (652)	2084 (620)	0.97
Carbohydrate (g/day)	252 (62)	236 (82)	231 (69)	0.67
Fat (g/day)	76 (24)	81 (34)	77 (24)	0.92
Protein (g/day)	98 (34)	148 (122)	111 (54)	0.37
Relative protein (g/kg)	1.4 (0.4)	1.9 (1.4)	1.4 (0.5)	0.42
Cognitive test 1	62 (15)	58 (12)	59 (12)	0.84
Cognitive test 2	51 (12)	55 (14)	53 (9)	0.83
Cognitive test 3	49 (15)	48 (6)	48 (6)	0.92

Abbreviations: BF = body fat; CR10 = 10 g of creatine per day group; CR20 = 20 g of creatine per day group; F = female, M = male; PLA = placebo group; VO_2_max = maximal oxygen consumption. Values are means (standard deviation).

**Table 2 brainsci-13-01276-t002:** Mean absolute change % (95% CI) from baseline to 6 weeks for cognitive performance and PFC O_2_Hb.

	CR10	CR20	PLA	Interaction *p*-Value
Cognitive test 1	3.6 (−3.4, 10.6)	19 (−0.19, 38.2)	18.7 (3.9, 33.5)	0.17
PFC O_2_Hb	−164.6 (−303.5, −25.7)	−100.3 (−217.1, 16.4)	71.7 (−154.5, 297.8)	0.07
Cognitive test 2	11.7 (−4.6, 27.9)	8.8 (−7.6, 25.3)	10 (−0.5, 20.5)	0.95
PFC O_2_Hb	−15 (−157.1, 127.2)	−25.9 (−92.8, 40.9)	81.6 (−162.1, 325.3)	0.50
Cognitive test 3	8.2 (−4.2, 20.7)	11.5 (3.8, 19.2)	4.9 (−4.8, 14.7)	0.59
PFC O_2_Hb	7.5 (−98, 113)	−41.1 (−84, 1.9)	−13.8 (−175.3, 147.8)	0.76

Abbreviations: CR10 = 10 g of creatine per day group; CR20 = 20 g of creatine per day group; O_2_Hb = oxygenated hemoglobin; PFC = prefrontal cortex; PLA = placebo group. Values are means (95% CI).

## Data Availability

The group data presented in this study are available upon request from the corresponding author. The individual data are not publicly available due to privacy and confidentiality.

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
