# Peer review of "Dose–Response of Creatine Supplementation on Cognitive Function in Healthy Young Adults"

_brainsci, 2023, doi:10.3390/brainsci13091276_

Round 1

Reviewer 1 Report

Thank you for the review invitation, indeed very important study work.

Few suggestions

- On the lack of brain creatin measurement, understandable that the authors addressed this as study limitation however it is also very important to elaborate this in the discussion on how with or without this measurement can bring different in study novelty and potential recommendation.

- On Figure 1. for me it seems not that readers friendly and would suggest to the authors to describe it in text or reformat it into table.

- Since this is an experimental study, suggesting the authors to mention the sampling method or sample calculation.

Author Response

Reviewer #1: Thank you for the review invitation, indeed very important study work.

Response: Thank you for the comment, and for the thorough review of the manuscript.

Few suggestions

- On the lack of brain creatine measurement, understandable that the authors addressed this as study limitation however it is also very important to elaborate this in the discussion on how with or without this measurement can bring different in study novelty and potential recommendation.

Response: Thank you for the comment. We have added the following sentences in the discussion section ‘One such study that did assess brain Cr levels (left dorsolateral PFC, left hippocampus, and occipital lobe) and cognitive performance (verbal learning and executive function) in healthy youth found that neither brain Cr levels nor cogni-tive performance were not influenced by the Cr supplementation [34]. Therefore, if ade-quate resources exist in future studies, Cr content should be measured in specific brain areas involved in expected cognitive functions (e.g., measurement of PFC Cr content along with assessment of executive function), allowing for the mapping of brain Cr and cognitive performance in the context of Cr supplementation.’.  

- On Figure 1. for me it seems not that readers friendly and would suggest to the authors to describe it in text or reformat it into table.

Response: Thank you for the comment but we think that Figure 1 is needed since it illustrates the trend for a decrease in oxygenated hemoglobin in the CR10 group during the first cognitive test. We do not think that written text has the same influence upon the reader.

- Since this is an experimental study, suggesting the authors to mention the sampling method or sample calculation.

Response: Thank you for the comment. The following information has been added to section 2.7 ‘Sample size was estimated based on long-term memory data from McMorris et al. [32] with effect size of d=0.81 and conducted assuming a power of 0.80 and alpha level of 0.05. It was estimated that a total of 21 subjects would be needed to locate a significant effect. Similar sample sizes have been used to show reduced mental fatigue and reduced cerebral oxygenated hemoglobin (creatine = 12, placebo = 12) [18].’.

Reviewer 2 Report

Dear Authors, I applaud your work and feel that you have conducted a great study that is methodologically sound with a thoughtful discussion. I just have some minor points you may wish to consider and comment on.

Introduction: Well written and highlight issues with older adults and other influences that may affect the brain. Less so with younger adults, but you do highlight that this area remains to be fully determined, and so provide a nice rational that is built throughout. Relevant literature included. 

One point to maybe add more: paragraph beginning line 51 to 61 - can you highlight the key difference between the studies more, what are the positives and negatives that you are taking and utilizing in your protocol i.e. at least 7 days with ~20 g/day, how do the studies influence your approach. 

Methodology: Again well written section. Detailed participant characteristic section/table. Within table 1, what is the unit for the cognitive tests? is it just AU? Was going to ask for more detail on validity and reliability, but see you have included relevant references to support this. But did you incorporate any familiarisation for these tests? Did you do any power calculation at all? Some results highlight trends, how many participants would have been ideal? Maybe worth adding as you do highlight a low sample size in the discussion too. 

Results: Straight forward section. I don't see any issues with your analysis and like that fact that you added some information on the thoughts of the participants on their condition at the end.  

Discussion:

Line 243, can you change the tense (we), and again on line 295. 

Line 279, with regards to the main limitation of the lack of understanding on the creatine levels in your participants - can you add some information from other studies that have measured this, and information on the expected increase with different strategies (if possible).

Line 296, amend 'may due to more efficient' to 'maybe due to a more efficient' 

Line 304, your second point for possible reasons of no cognitive improvement is the tests administered. I feel you could add more here. Is it a fact that the duration is not enough and we should focus more on longer tests, or the difficulty is not complexed enough to stress the cognitive function? Maybe the need to induce some physical fatigue is needed?? You reference one study that just employed one single test, but don't mention much around this, can you add more detail and suggest strategies moving forward perhaps?

Sorry for the limited feedback, but I really enjoyed your paper and think it is well constructed throughout.  

Author Response

Reviewer #2: Dear Authors, I applaud your work and feel that you have conducted a great study that is methodologically sound with a thoughtful discussion. I just have some minor points you may wish to consider and comment on.

Response: Thank you very much for the comment, and the overall positive review of the manuscript. 

Introduction: Well written and highlight issues with older adults and other influences that may affect the brain. Less so with younger adults, but you do highlight that this area remains to be fully determined, and so provide a nice rational that is built throughout. Relevant literature included.

One point to maybe add more: paragraph beginning line 51 to 61 - can you highlight the key difference between the studies more, what are the positives and negatives that you are taking and utilizing in your protocol i.e. at least 7 days with ~20 g/day, how do the studies influence your approach.

Response: Thank you. We feel that we mention that because of the inconsistencies between dosages and durations of previous studies (next paragraph from the lines you reference in your comment), we decided to investigate changes for the longest supplementation duration in a dose-response manner. That said, we have added the following sentence to the end of the paragraph you reference – ‘All in all, it appears that dosages 20 grams and over have shown the most consistent results at increasing brain Cr levels in healthy populations.’.

Methodology: Again well written section. Detailed participant characteristic section/table. Within table 1, what is the unit for the cognitive tests? is it just AU? Was going to ask for more detail on validity and reliability, but see you have included relevant references to support this. But did you incorporate any familiarisation for these tests? Did you do any power calculation at all? Some results highlight trends, how many participants would have been ideal? Maybe worth adding as you do highlight a low sample size in the discussion too.

Response: As regards the cognitive results, we mention in the second paragraph of section 2.3 that ‘A T-score of 50 indicates performance at the average level based on the input demo-graphic information of that participant.’. Each cognitive test we used does a standard practice or familiarization for each of the 3 tests. As regards the power calculation, we did complete a power calculation based on a 2007 McMorris et al. study and it was estimated that a total of 21 subjects was needed to locate a significant effect. We also mention that this sample size is similar across other studies investigating cognitive performance and cerebral oxygenated hemoglobin (12 in each group, creatine and placebo) (Watanabe et al., 2002).

‘Sample size was estimated based on long-term memory data from McMorris et al. [32] with effect size d=0.81 and conducted assuming a power of 0.80 and alpha level of 0.05. It was estimated that a total of 21 subjects would be needed to locate a significant effect. Similar sample sizes have been used to show reduced mental fatigue and re-duced cerebral oxygenated hemoglobin (creatine = 12, placebo = 12) [18].’

Results: Straight forward section. I don't see any issues with your analysis and like that fact that you added some information on the thoughts of the participants on their condition at the end. 

Response: Thank you, we appreciate this comment.

Discussion:

Line 243, can you change the tense (we), and again on line 295.

Response: Done.

Line 279, with regards to the main limitation of the lack of understanding on the creatine levels in your participants - can you add some information from other studies that have measured this, and information on the expected increase with different strategies (if possible).

Response: Thank you for the comment. Yes, we have added the following sentences in the discussion ‘One such study that did assess brain Cr levels (left dorsolateral PFC, left hippocampus, and occipital lobe) and cognitive performance (verbal learning and executive function) in healthy youth found that neither brain Cr levels nor cogni-tive performance were not influenced by the Cr supplementation [34]. Therefore, if ade-quate resources exist in future studies, Cr content should be measured in specific brain areas involved in expected cognitive functions (e.g., measurement of PFC Cr content along with assessment of executive function), allowing for the mapping of brain Cr and cognitive performance in the context of Cr supplementation.’.

Line 296, amend 'may due to more efficient' to 'maybe due to a more efficient'

Response: Changed to ‘maybe due to a more efficient’.

Line 304, your second point for possible reasons of no cognitive improvement is the tests administered. I feel you could add more here. Is it a fact that the duration is not enough and we should focus more on longer tests, or the difficulty is not complexed enough to stress the cognitive function? Maybe the need to induce some physical fatigue is needed?? You reference one study that just employed one single test, but don't mention much around this, can you add more detail and suggest strategies moving forward perhaps?

Response: Yes, thank you for the comment. We have added the following sentence ‘It may also be the case that a more complex set of cognitive tasks or ones that induce greater mental fatigue are needed to prompt additional brain activity in healthy young adults.’.

Sorry for the limited feedback, but I really enjoyed your paper and think it is well constructed throughout. 

Response: Thank you very much for the feedback you have given. It has definitely strengthened our manuscript. 

Reviewer 3 Report

was  the creatine level participants  measured before starting research protocol? using creatine monohydrate anyway article.

Author Response

Reviewer #3: was the creatine level participants measured before starting research protocol? using creatine monohydrate anyway article.

Response: No, creatine levels were not measured before starting the research study. We have addressed this in the discussion – ‘One such study that did assess brain Cr levels (left dorsolateral PFC, left hippocampus, and occipital lobe) and cognitive performance (verbal learning and executive function) in healthy youth found that neither brain Cr levels nor cogni-tive performance were not influenced by the Cr supplementation [34]. Therefore, if ade-quate resources exist in future studies, Cr content should be measured in specific brain areas involved in expected cognitive functions (e.g., measurement of PFC Cr content along with assessment of executive function), allowing for the mapping of brain Cr and cognitive performance in the context of Cr supplementation.’. 

Reviewer 4 Report

I read the manuscript entitled "Dose-response of creatine supplementation on cognitive function in healthy young adults" with great interest. The research design is appropriate, and the methods are adequately described. I have just a few comments.

- In materials and methods, it would be useful to provide more information on the randomization and blinding processes. For instance, how was blinding performed, or who is blinded? (See doi.org/10.3390/medicina57070647 and doi.org/10.1186/s13063-020-04607-5) If nobody was blinded, this should be added to the limitation.

- Regarding body composition evaluation, in particular fat mass, you should highlight that the gold standard for body composition evaluation (such as magnetic resonance for quantifying muscle mass or DXA for fat mass) has not been used to quantify those parameters, and results depend on BIA-based predictive equations with oscillating values depending on the formula applied and device (doi: 10.1017/S0007114522003749). It is clear that this is not your primary outcome, but the choice of the equation determines the validity of the FM% prediction (see here: doi.org/10.3390/nu15020278), and you could at least mention it.

- There is a difference in protein intake between CR10 and CR20 (1.4 vs. 1.9g/kg). Is it possible that a difference in diet (enriched in meat, for example) could influence body creatine stores and, in consequence, your results?

-Line 248: In line with your investigation, creatine supplementation (in a single administration) does not impact components of central fatigue during a task (doi.org/10.3389/fnut.2022.887523).

Overall, the article is interesting, well written, and organised.

Author Response

Reviewer #4: I read the manuscript entitled "Dose-response of creatine supplementation on cognitive function in healthy young adults" with great interest. The research design is appropriate, and the methods are adequately described. I have just a few comments.

Response: Thank you for the comment, and the overall positive review of the manuscript. 

- In materials and methods, it would be useful to provide more information on the randomization and blinding processes. For instance, how was blinding performed, or who is blinded? (See doi.org/10.3390/medicina57070647 and doi.org/10.1186/s13063-020-04607-5) If nobody was blinded, this should be added to the limitation.

Response: We have added some extra blinding information on lines 102-103 to now read - ‘This study was a double-blind, repeated measures study conducted in the Department of Kinesiology at the University of Northern Iowa (UNI).’ We have also added ‘and Randomization’ to section 2.4 title and added ‘to ensure all participants and investigators remained blinded throughout the study’ to lines 169-170.

- Regarding body composition evaluation, in particular fat mass, you should highlight that the gold standard for body composition evaluation (such as magnetic resonance for quantifying muscle mass or DXA for fat mass) has not been used to quantify those parameters, and results depend on BIA-based predictive equations with oscillating values depending on the formula applied and device (doi: 10.1017/S0007114522003749). It is clear that this is not your primary outcome, but the choice of the equation determines the validity of the FM% prediction (see here: doi.org/10.3390/nu15020278), and you could at least mention it.

Response: Thank you for this comment. We have added the following sentence to the end of the first paragraph of section 2.3 – ‘It is important to note that BIA is not the gold standard for body composition evaluation and although it is not a primary outcome in this study, the BIA results depend on BIA-based predictive equations with oscillating values depending on the formula applied and device [23].’.

- There is a difference in protein intake between CR10 and CR20 (1.4 vs. 1.9g/kg). Is it possible that a difference in diet (enriched in meat, for example) could influence body creatine stores and, in consequence, your results?

Response: Currently, research indicates that brain creatine increases in response to creatine monohydrate supplementation. When creatine monohydrate diffuses from the small intestine, the water molecule is gone by this point. Thus, the creatine molecule that remains is identical to what the liver produces. Dietary creatine from food is no different. Therefore, it is unlikely that the addition of nutrients such as carbohydrate or protein will have any effect on brain creatine uptake. Even if total protein was matched, there could have been differences between sources of protein (animal vs. plant) which may have a small impact on creatine intake and creatine levels in the brain. That said, there are also limitations with dietary food records as they represent a brief snapshot of what an individual is ingesting. We have added the following sentence in the limitations section – ‘Finally, although total protein was not matched in this investigation, the difference in protein intake is still challenging to quantify Cr content and is highly unlikely to have influenced the results.’

-Line 248: In line with your investigation, creatine supplementation (in a single administration) does not impact components of central fatigue during a task (doi.org/10.3389/fnut.2022.887523).

Response: Thank you for your comment. We feel that this paper is based upon an acute dose of creatine or creatine-based multi-ingredient pre-workout supplement and exercise performance and not related enough to the topic at hand (creatine, cognitive performance, and PFC oxygenation). Therefore, we aim to forgo referencing it in the manuscript.

Overall, the article is interesting, well written, and organised.

Response: Thank you very much.

Round 2

Reviewer 4 Report

Well done!